# Green and Efficient Extraction of Phenolic Components from Plants with Supramolecular Solvents: Experimental and Theoretical Studies

**DOI:** 10.3390/molecules29092067

**Published:** 2024-04-30

**Authors:** Bo-Hou Xia, Zhi-Lu Yu, Yu-Ai Lu, Shi-Jun Liu, Ya-Mei Li, Ming-Xia Xie, Li-Mei Lin

**Affiliations:** 1Key Laboratory for Quality Evaluation of Bulk Herbs of Hunan Province, Hunan University of Chinese Medicine, Changsha 410208, China; xiabohou@163.com (B.-H.X.); yuzhiluuu@163.com (Z.-L.Y.); luyuai@stu.hnucm.edu.cn (Y.-A.L.); lsj15574582159@163.com (S.-J.L.); liyamei163@163.com (Y.-M.L.); 2State Key Laboratory of Natural Medicines, China Pharmaceutical University, Nanjing 211198, China

**Keywords:** green extraction, supramolecular solvent, phenolic components, molecular dynamics simulation, confocal laser scanning microscopy, *Prunella vulgaris*

## Abstract

The supramolecular solvent (SUPRAS) has garnered significant attention as an innovative, efficient, and environmentally friendly solvent for the effective extraction and separation of bioactive compounds from natural resources. However, research on the use of a SUPRAS for the extraction of phenolic compounds from plants, which are highly valued in food products due to their exceptional antioxidant properties, remains scarce. The present study developed a green, ultra-sound-assisted SUPRAS method for the simultaneous determination of three phenolic acids in *Prunella vulgaris* using high-performance liquid chromatography (HPLC). The experimental parameters were meticulously optimized. The efficiency and antioxidant properties of the phenolic compounds obtained using different extraction methods were also compared. Under optimal conditions, the extraction efficiency of the SUPRAS, prepared with octanoic acid reverse micelles dispersed in ethanol–water, significantly exceeded that of conventional organic solvents. Moreover, the SUPRAS method demonstrated greater antioxidant capacity. Confocal laser scanning microscopy (CLSM) images revealed the spherical droplet structure of the SUPRAS, characterized by a well-defined circular fluorescence position, which coincided with the position of the phenolic acids. The phenolic acids were encapsulated within the SUPRAS droplets, indicating their efficient extraction capacity. Furthermore, molecular dynamics simulations combined with CLSM supported the proposed method’s mechanism and theoretically demonstrated the superior extraction performance of the SUPRAS. In contrast to conventional methods, the higher extraction efficiency of the SUPRAS can be attributed to the larger solvent contact surface area, the formation of more types of hydrogen bonds between the extractants and the supramolecular solvents, and stronger, more stable interaction forces. The results of the theoretical studies corroborate the experimental outcomes.

## 1. Introduction

Recent years have seen a surge in the exploration and utilization of natural bioactive compounds, sparking significant interest. This is because natural products serve as crucial sources in a variety of industries including cosmetics, food, and pharmaceuticals [1,2]. Phenolic compounds are a heterogeneous group of secondary metabolites widely distributed in plants among the bioactive phytochemicals [3]. They are acknowledged for their therapeutic properties and health benefits, including antioxidant, antibacterial, antiobesity, antidiabetic activities, and so on [4]. As for their numerous biological attributes, the antioxidant characteristics of phenolic compounds are responsible for a significant portion of their protective effects and are related to offering protection against oxidative-stress-related diseases like metabolic diseases, cancer, and other diseases [5]. Phenolic-rich plant extracts have become a commercial ingredient in food and nutritional products [6]. It is necessary to take them out of the raw material matrix to use them as medicinal agents or natural antioxidants. However, differences in the nature and chemical structure of phenolics, along with the presence of interfering compounds, can impact their solubility and separation properties. Thus, phenolic compounds are commonly extracted from plant materials in their crude state [7]. Therefore, there is a resurgence of interest in the development of efficient, green, and safe extraction processes of phenolic compounds, which will greatly improve their economic applications.

The initial and crucial step in the extraction processes (purification and isolation) of bioactive compounds from botanical sources is extraction. The traditional extraction methods applied for phenol recovery from plants include maceration, reflux, and soxhlet extraction, which are widely used today. Yet these techniques exhibit some specific drawbacks. For example, they are time-consuming, exhibit low extraction selectivity, and thermolabile compounds may decompose when a high temperature is used [8]. What is more, the conventional organic extraction solvents commonly used in these methods are flammable, non-degradable, or/and toxic. Their utilization is recognized to be detrimental to the environment and human health, and the extracts must still be processed before subsequent use [9,10]. In light of the emergence of the “green chemistry” ideology, it is crucial to substitute conventional organic solvents with environmentally friendly and secure alternatives [11,12]. Although water is recognized as the greenest and most harmless solvent, its efficiency in extracting bioactive substances from plants is very limited. The large amount of solvent is difficult to evaporate and concentrate, the extremely high temperature destroys the activity, and the extract’s complex composition necessitates further purification through columns [13,14]. Therefore, eco-friendly solvents including ionic liquids (ILs), deep eutectic solvents (DESs), and supercritical fluids like CO_2_ have been gradually investigated for the extraction and isolation of natural ingredients [15,16,17]. However, each of these solvents has its advantages and disadvantages. As for ionic liquids, they stand out due to their numerous composition options and capacity to modify the polarity of the target extract through changing solvent compositions, ranging from dipolar non-hydrogen-bonding solvents (DMF, DMSO, or acetonitrile) to polar hydrogen-bonding solvents (primary alcohols or water). Regrettably, the similar toxicity to organic solvents and the low biodegradability make the “green” character of ILs questionable [18]. Subsequently, DESs have been pioneered as a possible alternative to ILs. They have a greater extraction efficiency for polar chemicals because of the creation of intramolecular hydrogen bonds. Low preparation cost, low toxicity, and biodegradability also make DESs superior to conventional solvents [19,20]. The high viscosity of DESs at room temperature is one of their main drawbacks, hindering the transfer of solutes of interest and limiting their use in large-scale industry [21]. In addition, they may carry other biochemicals (e.g., proteins, polar lipids, sugar residues, cellulose, etc.) during the extraction process [22]. Supercritical fluids possess a low viscosity similar to gases and high diffusivity, allowing for the easy penetration of plant materials with rapid mass transfer rates. However, the primary drawback of utilizing CO_2_ in its supercritical state as a solvent is its limitation to pro-CO_2_ molecules. This restricts its applicability to small non-polar molecules due to CO_2_’s inherent non-polar nature [23].

It is essential to develop environmentally friendly, straightforward, and effective solvents for the extraction of bioactive compounds, such as phenols, from plant materials. These solvents should address the limitations of current extraction methods. SUPRASs are nanostructured liquids formed through the spontaneous self-assembly and co-coagulation of amphiphilic compounds in colloidal solutions [24]. The constituent molecules of SUPRASs’ ordered structure possess both hydrophilic and hydrophobic characteristics, with varying polar regions [25]. This molecular diversity allows SUPRASs to engage with low-molecular-weight solutes through multiple intermolecular forces during extraction, while excluding larger molecules such as polysaccharides and proteins through size exclusion and precipitation, respectively. Furthermore, different SUPRAS formulations have been developed to target specific bioactive compounds, like flavonoids and phenolic acids, by modifying the amphiphilic molecular composition and the cohesive environment [26,27].

In this study, the prepared octanoic acid–ethanol–water-based SUPRAS was applied to extract the main phenolic acids (caffeic acid, salviaflaside, and rosmarinic acid) from *Prunella vulgaris*, a medicinal and dietary plant. Depending on the yields of the target chemicals, a single factor was applied to optimize the conditions of SUPRAS extraction. Significantly, the extraction efficiency and antioxidant activity of SUPRAS extracts were evaluated against those obtained by conventional methods. The results indicate that SUPRAS extraction is a promising approach, offering superior extraction efficiency, more convenient extraction conditions, a simpler preparation method, and enhanced antioxidant capacity. Additionally, this study employed laser confocal microscopy to examine the microstructure and phenolic acid distribution post-extraction. Molecular dynamics simulations were also utilized to elucidate the intermolecular interactions between the extracted compounds and the extractant. This approach investigated the varying forces and mechanisms of action underlying the SUPRAS-mediated extraction of different bioactive compounds. The synergy of experimental and theoretical analyses underscores the potential of SUPRAS for the efficient and environmentally friendly extraction of bioactive ingredients.

## 2. Material and Methods

### 2.1. Chemical Reagents

n-Octanol (99% analytical grade, AR), n-octanoic acid (99.0% AR), n-decanol (98%), and n-decanoic acid (99%, AR) were obtained from Macklin Biochemical Co., Ltd., Shanghai, China. Absolute ethanol was provided by Sinopharm Chemical Reagent Co., Ltd., Shanghai, China. 2-Aminoethyl diphenylborinate (2-APB, 97%) was sourced from Sigma Aldrich, Merck, Germany. Reference standards, including caffeic acid and rosmarinic acid (both ≥98% purity), were purchased from Chengdu DesiTe Biological Technology Co. Ltd., Chengdu, China. The salviaflaside standard was prepared at the laboratory of Hunan University of Chinese Medicine. Methanol was purchased from TEDIA (Fairfield, OH, USA). Purified water was prepared using a Mili-Q water purification system (Millipore, Burlington, MA, USA). All other reagents were of analytical quality and acquired commercially.

### 2.2. Pretreatment of Prunella vulgaris

*Prunella vulgaris* (PV) was acquired from the GaoQiao Natural Herbal Special Market in Hunan Province, China, and confirmed as the dried fruit spikes of PV at Hunan University of Chinese Medicine. The herb specimens were dried until a consistent weight was achieved, crushed, sifted through 40-mesh screens, and placed in a desiccator for storage prior to further experiments.

### 2.3. SUPRAS Synthesis and Characterization

A variety of SUPRASs were produced by dissolving alkanols/alkanoic acids with long chains in varying percentages of ethanol, followed by the addition of water (pH ~3) to stimulate the formation of the amphiphile aggregates. The total volume of the mixture containing the three components was maintained at 50 mL. The mixtures were shaken on a vortex and then centrifuged (TGL-20MB high-speed refrigerated centrifuge, Changsha Xiangzhi Centrifuge Instruments Co., Ltd., Changsha, China) for 10 min at 5000 rpm. Following centrifugation, a fresh upper liquid phase referred to as SUPRAS was isolated, which reached equilibrium with the rest of the solution. The two distinct phases (SUPRAS and equilibrium solution, EqS) were collected separately and stored in sealed glass containers at room temperature (~20–25 °C) for up to one week. The production process for SUPRAS is illustrated in Figure 1.

The makeup of the SUPRAS relied on the composition of amphiphiles and the proportion of water to ethanol. To further understand the properties of the SUPRAS, it was necessary to determine the composition of the SUPRAS. Weight percentages (*w*/*w*, %) were used to measure the composition of amphiphiles, ethanol, and water in the SUPRAS. The alcohol meter was used to calculate the ethanol percentage. The amphiphile content was weighed after ethanol and water were evaporated, and the water content was determined by weight difference.

### 2.4. SUPRAS-Based Extraction and Optimization of Phenolic Compounds from PV

The entire extraction procedure of phenolic acid is shown in Figure 1. Extractions were performed in a 2.0 mL microfuge tube by mixing 150.00 mg of crushed PV with 0.4 mL SUPRAS and corresponding EqS 0.8 mL. The different types of SUPRASs were previously produced according to the method described in Section 2.3. Following this, the blends were agitated at 3000 rotations per minute for 30 s and then subjected to ultrasonic extraction for 10 min. Subsequently, the combinations underwent centrifugation at 10,000 revolutions per minute for 10 min to ensure complete elimination of any remaining residues from the solvents. To conclude, the resulting liquid supernatants were gathered and thinned with pure methanol prior to conducting HPLC assessment.

The extraction yields of caffeic acid, salviaflaside, and rosmarinic acid from PV were chosen as standards for valorization. Through one-factor-at-a-time experiments, variables of different solvent compositions and extraction conditions were optimized. Initially, a study was conducted on the influence of amphiphiles (such as octanol, octanoic acid, decanol, or decanoic acid), and the SUPRAS displaying the optimal production of the desired compounds was chosen.

Next, the ethanol concentration (from 20% to 35%, *v*/*v*) and the pH (from 1 to 5) for SUPRAS formation were studied. Furthermore, the influence of the ratios of SUPRAS:EqS (*v*/*v*, from 1:2 to 1), the volume of extraction solvent (from 0.6 to 1.8 mL), and ultrasonic time (from 5 to 45 min) were assessed to determine the most effective extraction yield. All experiments were conducted at room temperature (~25 °C). Likewise, the ultrasonic-assisted traditional organic reagent (ethanol) extraction method, methanol reflux extraction method, and DES-based extraction approach were also used to extract PV under optimal conditions, which provided a reference and comparison for the SUPRAS extraction process.

### 2.5. Determination of Phenolic Compounds by High-Performance Liquid Chromatography (HPLC)

Phenolic compounds were analyzed using an Agilent 1260 reversed-phase HPLC system (Agilent, Santa Clara, CA, USA) equipped with a ZORBAX SB-C18 column (4.6 × 250 mm, 5.0 μm). To identify the compounds, the retention time of the standards was compared with the target substances (Figure 2). The mobile phase consisted of 0.1% formic acid in water as phase A and methanol as phase B. The column was maintained at 25 °C, with a flow rate of 1.0 mL·min^−1^ and an injection volume of 10 μL. The detection wavelength for phenolic acids was set at 330 nm. The gradient elution was performed under the following conditions: 0–10 min, gradient from 5% to 35% B; 10–40 min, gradient from 35% to 37% B; 40–45 min, constant at 37% B. Prior to HPLC analysis, all samples were filtered through a 0.22 μm filter.

Related studies have shown that the alkyl carboxylic acids used for the preparation of SUPRASs as extractants do not affect the quality of the detection of analytes in HPLC analysis [28]. Therefore, the use of HPLC is reliable for the analysis in this study.

### 2.6. Antioxidant Activity Assays of Different Extraction Methods

The antioxidant properties of extracts obtained using the ideal SUPRAS conditions were assessed through DPPH and ABTS•^+^ assays, along with other techniques. The half-maximal inhibitory concentration (*IC*_50_) was calculated for comparison [29].

#### 2.6.1. DPPH Radical Scavenging Activity Assay

The DPPH radical scavenging activity of the extracts was assessed using a spectrophotometric method based on the DPPH assay, as previously described, with slight modifications [30]. Samples prepared by SUPRAS and other extraction methods were diluted to five concentrations. Then, 1.0 mL of a 0.09% (*w*/*v*) DPPH solution in methanol was added to a 0.25 mL sample solution. After incubating at 25 °C for 30 min without light, the absorbance was tested at 517 nm. The blank for the DPPH solutions was 1.0 mL of methanol, and no antioxidants were present in the control group. The formula for assessing the effectiveness of the DPPH radical scavenging was as follows.
DPPH radical scavenging effect (%) = [A_control_ − (A_sample_ − A_blank_)/A_control_] × 100%

#### 2.6.2. ABTS•^+^ Radical Scavenging Activity Assay

The activity of scavenging ABTS•^+^ radicals was determined using the ABTS•^+^ rapid method included in the total antioxidant capacity test kit. ABTS•^+^ radical scavenging activity was measured according to the total antioxidant capacity test kit (ABTS•^+^ rapid method). Samples prepared by SUPRAS and other extraction methods were diluted into five concentrations and then measured according to the instructions. Following gentle shaking of the mixtures and 6 min of incubation at 25 °C, the absorbance at 414 nm was quantified using an enzyme-linked analyzer (BioTek Synergy Multifunctional Microplate Assay, Hunan Zidonglai Biotechnology Co., Ltd., Changsha, China). The antioxidant capabilities of the extracts are represented as *IC*_50_. The scavenging effects against ABTS•^+^ radicals were calculated using the formula:ABTS•^+^ radical scavenging effect (%) = [(A_blank_ − A_sample_)/A_blank_] × 100%

### 2.7. Characterization of Droplet Microstructure and Target Compound Interactions by Confocal Laser Scanning Microscopy (CLSM)

To validate and observe the unique microstructure of droplets related to SUPRASs and their interaction with phenolic acid compounds, an analysis was conducted utilizing confocal laser scanning microscopy (CLSM). The extracts obtained through various extraction methods were diluted with anhydrous methanol to match the concentration of the SUPRAS extracts. All samples were treated with 0.25% (*w*/*v*) 2-aminoethyl diphenylborinate (2-APB) in anhydrous methanol for 15 min and then centrifuged at 10,000 rpm for 10 min to eliminate impurities. The entire process was conducted in the absence of light. A drop of 5 μL of the 2-APB mixture was placed on a microscope slide, covered with a 22 × 22 mm coverslip, and sealed. CLSM fluorescence images were captured using excitation wavelengths of 405 and 488 nm and emission wavelengths of 450 and 525 nm to examine the phenolic acids’ extraction conditions and their spatial distribution. Additionally, images under white light conditions were obtained to depict the microstructure of SUPRAS droplets. The settings and parameters for fluorescence were consistently applied across all images and channels. The fluorescence images were compared with the overlapping images of the SUPRAS, thus demonstrating that the droplet structure of the SUPRAS could effectively extract phenolic acids.

### 2.8. Molecular Dynamics Simulation Analysis

The analysis using the GROMACS 2020.6 package and Amber (99SB-ildn) force field included a molecular dynamics simulation [31]. The topological information of molecules, such as ethanol, octanoic acid, water, and rosmarinic acid was generated by the LigParGen. The Multiwfn-calculated partial charges were determined based on restrained electrostatic potential (RESP) charge [32].

Three extraction mechanisms were all created within the cubic box with dimensions of 10 nm × 10 nm × 10 nm using GROMACS. More specifically, the SUPRAS container comprised 700 molecules of octanoic acid, 1500 ethanol molecules, 3250 water molecules, and 100 molecules of rosmarinic acid. The ethanol solution (70%, *v*/*v*) consisted of 3600 ethanol molecules, 4701 water molecules, and 100 molecules of rosmarinic acid. The water compartment was composed of 15,274 water molecules and 100 molecules of rosmarinic acid. A cut-off strategy with a distance cut-off of 1.0 nm was utilized for both van der Waals and short-range Coulomb interactions. The system’s pressure was maintained at 1 atm using the Parrinello–Rahman technique during the production stage and the Berendsen approach during the stabilization phase.

The temperature was maintained at 298 K throughout the whole process. Periodic boundary conditions were applied in all directions, and the generating phase lasted for 100 ns with a step value of 2 fs. Lastly, the simulated processes were analyzed for the hydrogen bonding interaction, average non-covalent interaction, etc.

### 2.9. Statistical Analysis

Every trial was conducted in triplicate, and the findings were presented as average ± SD. Utilizing SPSS statistics 17.0, the statistical assessments of one-way ANOVA and Tukey’s examinations were performed to evaluate the notable distinctions within the experimental information. A significance level of *p* < 0.05 was deemed statistically important.

## 3. Results and Discussion

### 3.1. SUPRAS Production and Composition

SUPRAS are nanostructured liquids formed within amphiphile colloidal suspensions through a process of self-assembly and coacervation [25]. The use of SUPRASs aims to emphasize their characteristics in the formation process, taking into account the non-covalent interactions that hold the molecules together and the self-assembly processes of their synthesis [26]. The process of creating SUPRASs typically goes through two steps. Initially, amphiphiles are dissolved in either protonic or non-protonic dipolar solvents, leading to the self-aggregation of the colloidal system into vesicles or reverse micelles upon exceeding the critical aggregation concentration. Subsequently, the introduction of water (which acts as the “poor solvent” for the amphiphilic compounds) induces the aggregates to self-assemble into the SUPRAS, a novel densely packed phase with an inverted hexagonal configuration. Within this configuration, the hydrocarbon chains are dispersed in the organic solvent, while the carboxylic groups encircle the aqueous cavities.

For this research, ethanol was chosen as the primary ingredient for creating SUPRAS due to its reduced harmfulness. In fact, the octanol, decanoic acid, ethanol, decanol, and octanoic acid utilized in the creation process were classified as Generally Recognized as Safe (GRAS). The quantity of amphiphiles such as octanoic acid was fixed at a 5% volume-to-volume (*v*/*v*) ratio, as past research indicated that the volume of SUPRAS showed a direct correlation with the percentage of amphiphiles when the organic solvent ratio remained constant. Additionally, higher levels of octanoic acid necessitated an increased amount of ethanol, prompting the selection of the optimal 5% ratio for experimentation [33,34].

Moreover, the SUPRAS was prepared by centrifugation at 5000 rpm for 10 min in our experiments. The available experiments showed that the equilibrium phase separation conditions were reached after centrifugation at 3000 rpm for 5 min. In contrast, sonication at 40 kHz could not completely separate the two phases after 2 h [28]. These findings led to the recommendation of centrifugation for the production of the SUPRAS. Based on all the preliminary research above, the SUPRAS will be better used to extract bioactive in PV and it provides suitable conditions for further process optimizations.

### 3.2. Optimized Results of SUPRAS-Based Extraction of Phenolic Acid

The optimization procedure has been illustrated in Section 2.4 and the extraction efficiency of the phenolic acids was evaluated.

#### 3.2.1. Different Amphiphilic Organic Solvents

Selecting the appropriate amphiphiles was the primary consideration in optimizing the extraction process based on SUPRASs. The composition and properties of the amphiphiles play a key role in the efficiency of extracting the desired bioactive components [35]. Through careful selection of amphiphiles and the coacervation environment, these molecules can assemble themselves to form nanostructured fluids with customizable properties. To achieve this, a combination of octanoic acid, octanol, decanoic acid, and decanol was used to create various SUPRASs. The remaining parameters were kept constant at 35% ethanol, pH~3 water, a ratio of SURAS: EqS of 1:2, and ultrasound treatment for 15 min at 25 °C. As shown in Figure 3A, the octanoic-acid-based SUPRAS extracted 14.40-fold more salviaflaside (0.216 mg·g^−1^) than decanol (0.015 mg·g^−1^) and 6.17-fold more than octanol (6.94 mg·g^−1^). The extraction yield of rosmarinic acid reached 1.126 mg·g^−1^, which was 3.06-fold more than that of decanoic acid, 14.25-fold more than that of decanol, and 3.42-fold more than that of octanol. The SUPRAS prepared with octanoic acid was found to be the most suitable for the extraction of phenolic acids.

This result might be explained as follows: Hydrogen bonding and dispersion played a significant role in influencing the extraction of target components by SUPRASs. As the hydrocarbon chain lengthened, hydrogen bonding decreased while the dispersion increased. In the case of polar compounds, hydrogen bonding served as an effective retention mechanism, leading to a preference for shorter-chain alcohols/alkyl carboxylic acids as they exhibited superior proton-donating capabilities compared to their longer-chain counterparts [35]. When comparing acids and alcohols, alkyl carboxylic acids were identified as better proton donors than alkyl alcohols [28]. Given the high polarity and the abundance of hydrogen bond acceptors in the target bioactivities, it was reasonable to anticipate that extraction would be optimized with the SUPRAS containing octanoic acid, as this compound was capable of forming stronger hydrogen bonds.

#### 3.2.2. The Percentage of Ethanol

Secondly, the impact of the different ethanol contents in the synthesis of SUPRASs was investigated. It was determined that when the proportion of ethanol in the solution increased, both the volume of the SUPRASs and the size of the spherical droplets forming them increased.

Next, the study explored how varying ethanol levels affect the production of SUPRASs. The results revealed that as the ethanol concentration rose, the size of the resulting spherical droplets and the overall volume of the SUPRASs also increased [35]. Thereby, the extraction efficiency can be affected and optimized. SUPRASs with different compositions and vesicle sizes used in the following experiments were generated by varying the relative ratios of water and ethanol [36]. The influence of ethanol content was investigated within 20% to 35% *v*/*v*, where the SUPRAS was stratified (Figure 3B).

As the percentage of ethanol increased, the extraction rate of caffeic acid did not change much, but the extraction rate of salviaflaside and rosmarinic acid increased significantly. The optimal extraction yield of phenolic acids was achieved at a solvent concentration of 35% (*v*/*v*). Phenolic acids, as highly polar compounds, were found to have higher extraction rates in SUPRASs with larger water cavities formed by higher ethanol percentages [37].

#### 3.2.3. pH

pH was identified as a critical factor influencing extraction efficiency due to its impact on analyte solubility and the charge density of the extracts and solvents [38]. The pKa values of caffeic acid and rosmarinic acid are 4.58 and 2.78, respectively. Both two phenolic compounds have low pKa values and are more acidic. To examine the influence of pH on the extraction performance, the pH of the water used in the synthesis of SUPRASs was systematically adjusted over a range of 1 to 5 using HCl. According to the results in Figure 3C, it is clear that the change in pH had a large effect on the extraction rate of phenolic acids. When the pH was 1, the extraction yield of rosmarinic acid reached 1.33 mg·g^−1^, ~1.43-fold higher than other values. This phenomenon accounts for the fact that phenolic compounds tend to take on molecular forms under acidic conditions (pH < pKa), which enables the target compounds to be easily extracted from the sample phase into the extraction phase [39].

#### 3.2.4. The Ratio of SUPRAS:EqS

Subsequently, the effect of the SUPRAS:EqS ratio on the extraction yields was investigated. There are two common strategies for utilizing SUPRASs in extracting target compounds from solid materials. The initial approach includes directly incorporating SUPRASs into the solid samples [8]. This method is recommended for extracting compounds that cover a broad range of polarity. The second strategy involves volumes of both SUPRASs and EqS being used [40]. The EqS is commonly used with the SUPRAS to wet the samples.

In this optimization, the total volume of solvents (SUPRAS + EqS) used for extraction was kept constant at 1.2 mL and the SUPRAS phase content ranged from 33% to 100%. Figure 3D demonstrates that when the SUPRAS phase increased, the extraction of the three phenolic acids increased considerably. This was possibly caused by the highly polar target compounds’ dispersion distribution between the SUPRAS and EqS phases [41]. The best option for maximizing the extraction efficiency was 100% SUPRAS.

#### 3.2.5. The Sample: Extractant Phase Ratios

Subsequently, we examined the effects of the extractant phase ratios (g:mL) ranging from 1:4 to 1:12 on the sample extraction (Figure 3E). For this purpose, the amount of PV powder was maintained at 150.00 mg and the volume of the SUPRAS varied from 0.6 to 1.8 mL. For the phenolic acids, the extraction rates grew gradually as the sample volume increased, reaching saturation at a ratio of 1:10 (g:mL). The extraction yields did not continue to improve with increasing solvent volume and even decreased slightly at 1:12 (g:mL) due to the decrease in mass transfer efficiency. The phase ratio of 1:10 (g:mL) was identified as the optimal choice for the extraction process.

#### 3.2.6. Ultrasonic Time

The extraction yields of target analytes were significantly influenced by ultrasonic time. During ultrasonic-assisted extraction, the ultrasonic wave ruptured the cell wall of the PV, allowing the vesicle solution to penetrate the cell and engage with the active ingredients [42]. To obtain the best extraction yields, different extraction times (5–45 min) were assessed under the same extraction conditions. As Figure 3F reveals, the extraction of phenolic acids was not impacted by the extraction time. The extraction process was finished in 25 min. SUPRASs exhibited high efficiency in extracting bioactive compounds from solid samples without the need for increased temperatures or longer heating durations in comparison to the conventional method of organic solvent reflux extraction. Under the optimal conditions, we further detected the content of total phenolic acids and found that the content was as high as 72.44%.

One of the most frequently utilized strategies in experimental studies is single-factor optimization to acquire initial optimal protocols. Nevertheless, our understanding also included the potential interactions among these factors. Subsequent investigations will involve a multiple regression analysis of the data to establish the optimal extraction conditions and examine the correlation between responses and variables through a quadratic ANOVA model. Additionally, RSM will be employed to effectively demonstrate the interactions among the factors impacting the extraction process.

### 3.3. Method Validation

The method validation included the standard curve, correlation coefficient (R^2^), relative standard deviation (RSD), stability, spiked recovery, LODs, and LOQs (as shown in Table 1). In this experiment, the standard curves were established for the qualitative and quantitative determination by HPLC. The correlation coefficients of the three phenolic acids were above 0.9996, confirming the excellent linearity. Next, under ideal conditions, six extractions were simultaneously carried out to evaluate the accuracy of the technique. The relative standard deviation (RSD) for caffeic acid, salviaflaside, and rosmarinic acid was found to be 3.31%, 3.14%, and 1.15%, in that order.

The limits of determination (LODs) of the three phenolic acids ranged from 0.08–0.16 μg mL^−1^ and the limits of determination (LOQs) ranged from 0.29 to 0.53 μg mL^−1^, determined using signal-to-noise (S/N) ratios of 3 and 10, respectively. To verify the method’s reliability, spiked recoveries were assessed by introducing varying standard concentrations into the sample solutions, and the results ranged from 100.81% to 101.98% with an RSD of 1.02–1.39%. Based on the above results, the extraction method established in this experiment was sufficiently accurate and reliable. It has potential applications in the determination of other pharmaceutical products.

### 3.4. Comparison of the Extraction Efficiency of Phenolic Acids by Different Extraction Methods

The choice of appropriate solvents was essential in the extraction chemistry. The extraction efficiencies of caffeic acid, salviaflaside, and rosmarinic acid obtained using SUPRASs were compared to those extracted by other different solvents previously applied. As shown in Table 2, the SUPRAS extracts of caffeic acid (0.240 mg·g^−1^) and salviaflaside (1.443 mg·g^−1^) were higher than those of the 30% ethanol method (0.146 mg·g^−1^ and 0.800 mg·g^−1^) and the DES method (0.213 mg·g^−1^ and 1.199 mg·g^−1^).

Similarly, SUPRASs showed the highest extraction yield of rosmarinic acid, which was ~1.78-fold more efficient (5.254 mg·g^−1^) than 30% methanol (2.957 mg·g^−1^), and it was higher than both the anhydrous methanol reflux method (4.667 mg·g^−1^) and the DES method (3.706 mg·g^−1^). The 0ANOVA analysis of the data further demonstrated that the phenolic acid yields extracted using SUPRASs were statistically significantly different (*p* < 0.05) compared to the other methods.

In addition, pretreatment, high temperatures, and large solvent volumes have been reported to obtain higher extraction rates. For example, the conventional solvent extraction method based on aqueous methanol had a solid-to-liquid ratio of 20 mL·g^−1^ and required two continuous refluxes for 2 h, which was time-consuming and energy-intensive [15]. Among the extraction methods assisted by ultrasound, the method of Huafu Wang et al. (2004) had a solid-to-liquid ratio as high as 500 mL·g^−1^ [43]. The method using DES, although reporting a lower solvent consumption of 10 mL·g^−1^, required 80 °C heating and stirring for 46 min [44]. Hence, considering the solvent dosage, the duration of extraction, and the energy utilized, the SUPRASs demonstrate significant potential in extracting bioactive compounds.

### 3.5. Comparison of Antioxidant Activity by Different Extraction Methods and Solvents In Vitro

Phenolic compounds, such as caffeic acid, salviafaside, and rosmarinic acid, are the primary factors behind PV’s antioxidant characteristics [45]. Previous studies indicated that rosmarinic acid and caffeic acid possess excellent antioxidant capabilities [46,47]. Therefore, examining extraction methods is crucial for assessing the antioxidant potential of various solvent extracts. The assessment of plant extract antioxidant activity was carried out using the DPPH and ABTS•^+^ assays in this study.

According to the method in Section 3.4, extraction using SUPRASs was compared in terms of antioxidant properties with 30% ethanol, anhydrous methanol, and DES (36% vol water in ChCl/ethylene glycol). The results (Figure 4) showed that the extraction solvents also had a significant influence on the antioxidant properties of the extracts. According to the ABTS•^+^ radical scavenging assays, the extracts prepared with innovative green solvents like SUPRASs (*IC*_50_ = 2.061 mg·mL^−1^) and DESs (*IC*_50_ = 2.001 mg·mL^−1^) had significantly higher antioxidant capacities than those prepared with conventional organic solvents (*IC*_50_ = 2.386 mg·mL^−1^ for 30% ethanol, and 3.262 mg·mL^−1^ for methanol). Although the *IC*_50_ value of the DES extracts was slightly lower than that of the SUPRAS extracts, the inhibition rate of the SUPRAS extracts was higher than that of the DESs when the concentration exceeded 1 mg·mL^−1^. As for the DPPH assays, the results showed that the DPPH• generation was inhibited by all the extracts with the following degree of inhibition: SUPRAS > DES > 30% ethanol > methanol (*IC*_50_ = 1.787, 1.997, 2.049, and 2.615 mg·mL^−1^, respectively). The results suggested that the SUPRAS extracts exerted the highest free radical scavenging capacity. Therefore, SUPRASs, which also exhibit good antioxidant potential, could be used as reliable and effective extraction solvents.

### 3.6. CLSM Results for Efficient Extraction Performance of SUPRAS

Previous studies reported that SUPRAS comprise round droplets distributed in a continuous phase, which is also seen as a characteristic feature of SUPRASs that distinguish them from ordinary solvents and is thought to be related to their superior extraction ability [28]. Under achromatic irradiation conditions of CLSM, we observed the spherical droplet structure of SUPRASs, while similar structures were completely absent in other solvents (Figure 5). However, the droplet structure of the SUPRAS itself has been studied several times, but its interaction with the extracts has not been mentioned explicitly. To gain deeper insight into the extraction procedure and explore the extraction mechanism, CLSM was used in this experiment to concurrently capture pictures of the solvent microstructures and the distribution of phenolic acid after the extraction.

To determine the location and morphology at the CLSM interface, the target compound (e.g., phenolic compounds) must be fluorescent or treated with a fluorescent dye so that a clear signal is available [48]. In this experiment, 2-APB was applied to amplify the fluorescence signal to visualize the phenolic acid compounds.

As indicated in Figure 5, phenolic acids exhibited blue fluorescence when excited at 450 nm and emitted at 405 nm and displayed green fluorescence when excited at 488 nm and emitted at 525 nm. The fluorescence positions of phenolic acids in the CLSM images matched the positions of the SUPRAS droplets, indicating the encapsulation of phenolic acids in vesicles by the SUPRAS droplets. The fluorescence images of the SUPRAS extracts (Figure 5A) appeared more concentrated and intense compared to those of the other solvents, while the fluorescence from other solvent extracts (Figure 5B–D) was weak and scattered. This confirms that the SUPRAS effectively extracted phenolic acids into its droplets. The ability of SUPRAS droplets to enrich and efficiently extract the target compounds was evident. Visualizing the specifics of the droplets and extracts following SUPRAS extraction aids in a more direct examination of the extraction mechanism.

### 3.7. Molecular Dynamics Mechanism for Efficient Extraction Performance of SUPRAS

The simulation parameters chosen for the molecular dynamics simulations were based on the actual content of the solvents and extracts used in the experiments. To further ensure the realism and reliability of the simulations, the composition of the supramolecular solvents was determined to facilitate the determination of the number of solvent molecules in the simulations. After optimization, the composition of the SUPRAS was found to consist of 43.96% octanoic acid, 30.33% ethanol, and 25.17% water. The substantial presence of amphoteric solvents enhanced the extraction efficiency of the SUPRAS.

In general, the hydrogen bonds developed between the solvent and phenolic compounds were extremely effective retention mechanisms of extraction and promoted the dissolution of the extractive [37]. Molecular dynamics simulations were applied to understand the extraction mechanism of the SUPRAS, which had high extraction efficiency in comparison to other solvents. In this study, concerning the existing extraction methods, three typical solvents—the octanoic acid–ethanol–water SUPRAS, ethanol (30%, *v*/*v*), and water—were selected for comparison, and rosmarinic acid, which had the highest extraction amount, was selected as the typical phenolic compound from PV for subsequent simulations.

The distribution of rosmarinic acid molecules in three distinct solvents was initially simulated, as depicted in Figure 6. It was observed that the rosmarinic acid molecules formed a compact cluster in water, whereas they were more dispersed in the ethanol system. The solvent-accessible surface area (SASA) plays a crucial role in solubilization and extraction efficiency [49]. Within the initial 20 ns of the simulation, the SASA value of rosmarinic acid molecules in water decreased significantly, whereas it decreased at a slower rate and remained relatively constant in both the ethanol and SUPRAS. The SASA values, as shown in Figure 7B, indicate the stable dissolution state of rosmarinic acid molecules in the SUPRAS across the three solvents. The SASA of the rosmarinic acid molecule in the SUPRAS was 348.79 nm^2^ and in ethanol was 367.91 nm^2^, both of which were much higher than that of water at 167.37 nm^2^.

By combining the simulation results of the distribution state and the SASA values, an intriguing phenomenon was observed: the SASA values of rosmarinic acid molecules in the SUPRAS and the ethanol system were similar, yet the molecules in the SUPRAS exhibited a notable aggregation. This suggests that SUPRASs may possess an enrichment effect for bioactive substances, even when extracting with equal efficiency. Consequently, SUPRASs are promising candidates for further exploration and research in the field of rapid, efficient, and micro-extraction methods.

Furthermore, the mean duration of hydrogen bonding among solvents and reactants might reflect the strength of the hydrogen bonds connecting hydrogen donors and acceptors. This is employed as a key parameter in molecular dynamics simulations to quantify the binding strength between the solvent and the reactant [50]. In general, stronger interactions between solvent and reactants could translate into lower transition-state free energies, thus increasing the reaction rate and extraction efficiency. As shown in Figure 7E, In the SUPRAS, rosmarinic acid exhibited an average hydrogen bond lifetime of 2.11 ps, surpassing that observed in ethanol (1.92 ps) and water (1.52 ps) systems. This indicates that the hydrogen bond stability of rosmarinic acid in the SUPRAS was superior to that in the other solvents, contributing to the increased extraction efficiency and solubility of phenolic acids.

The average non-covalent interaction (aNCI) was utilized to thoroughly examine the feeble interactions in the ever-changing conditions [51]. The locations and variety of non-covalent interaction sites in SUPRAS systems could be visually illustrated [52]. As seen in Figure 8A, there are more dark blue areas of hydrogen bonding between the solvent and rosmarinic acid molecules in the SUPRAS system. Compared to the water or ethanol systems, the SUPRAS solvent system contained more solvent–target molecule interactions (reflected in the red dashed line). However, in the ethanol system, there was more solvent–solvent hydrogen bonding (reflected by the blue dashed line). Thus, even though the water system contained somewhat more hydrogen bonds in total, the SUPRAS was more exclusive and efficient in extracting rosmarinic acid.

The thermal fluctuation index (TFI) could indicate the stability of the interaction mentioned in the aNCI analysis. The degree of stability of these interactions varied, ranging from highly stable to very weak, as evidenced by the color calibration transitioning from blue to red. Through an analysis of the TFI plots, it was observed that the red region representing rosmarinic acid molecules in the SUPRAS system was notably smaller compared to the other systems. Consequently, the hydrogen bonding between phenolic acids and solvent molecules in the SUPRAS system exhibited greater stability, facilitating the extraction process.

In recent developments, molecular dynamics analysis, in conjunction with quantum chemical calculations, has been employed to confirm and quantitatively assess the interactions between extractants (e.g., NADES) and various components [53,54]. In this study, an attempt was made to further reveal the extraction of the SUPRAS through molecular dynamics simulations, hoping to provide insight into the efficient extraction using SUPRASs. Also, it was theoretically demonstrated that the SUPRAS had an efficient extraction capacity.

### 3.8. Existing Problems and Prospects

However, drawbacks still exist with SUPRASs. For instance, further purification processes are necessary to separate bioactivities, and exploring how to recover and reuse SUPRASs is an ongoing area of research. Conversely, the limited industrial application of SUPRASs is attributed to insufficient application knowledge and a lack of research on diverse types of SUPRASs. Moving forward, more specific and appropriate SUPRASs should be developed, researched, and utilized. Enhanced characterization techniques will be employed to understand the structure formation and extraction mechanisms of various SUPRASs, ultimately enhancing their utility across different industries. This potential advancement positions SUPRASs to potentially replace traditional organic reagents for extracting and enriching target compounds.

## 4. Conclusions

This study introduced a new, environmentally friendly, and effective approach using an octanoic acid–ethanol–water-based SUPRAS for extracting phenolic compounds from PV. The solvents used in this method are green and safe, which when combined with the outstanding antioxidant capacity of phenols, makes them more suitable to be extracted from plants as active ingredients in food or nutritional products. Under the optimal extraction conditions, the extraction yields of caffeic acid, salviaflaside, and rosmarinic acid reached 0.240 mg·g^−1^, 1.443 mg·g^−1^, and 5.254 mg·g^−1^, respectively. In comparison to other documented methods of extraction, PV extractions utilizing SUPRASs demonstrated the highest yields and antioxidant activities. HPLC analysis indicated that the SUPRAS extract exhibited specific characteristics with minimal impurities. Additionally, the extraction mechanism and interactions between the target molecules and the SUPRAS were investigated using CLSM characterization and molecular dynamics simulations. Compared to alternative solvents, SUPRASs possess the capability to extract phenolic compounds into a droplet structure. The enhanced stability of hydrogen bonding and larger solvent-accessible surface area of SUPRASs could elucidate the variances in efficiency among solvents and enhance our understanding of the extraction process. In light of these findings, the tailored SUPRAS solvent presents as an eco-friendly and effective option for extracting natural products essential in the food, pharmaceutical, and chemical sectors.

## Figures and Tables

**Figure 1 molecules-29-02067-f001:**
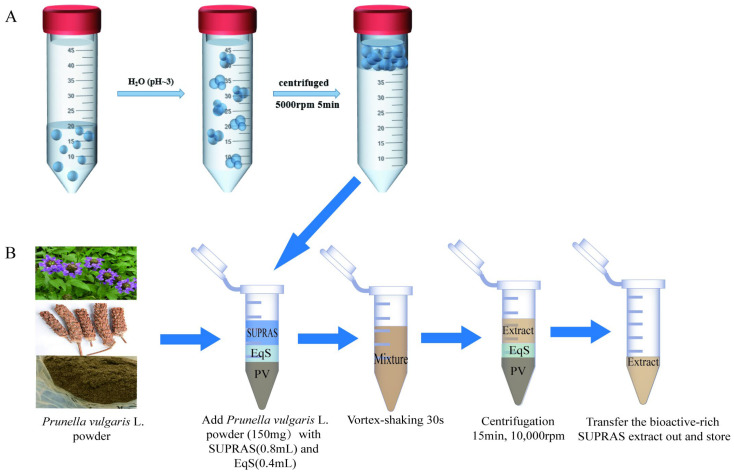
Schematic diagram showing the synthesis of SUPRAS (**A**) and the extraction process of PV (**B**).

**Figure 2 molecules-29-02067-f002:**
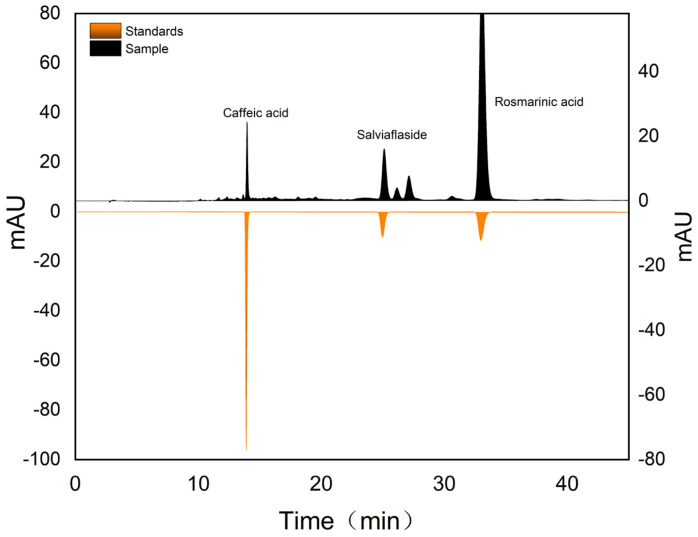
HPLC chromatograms of SUPRAS extraction of PV and three standards were detected and the retention time was compared.

**Figure 3 molecules-29-02067-f003:**
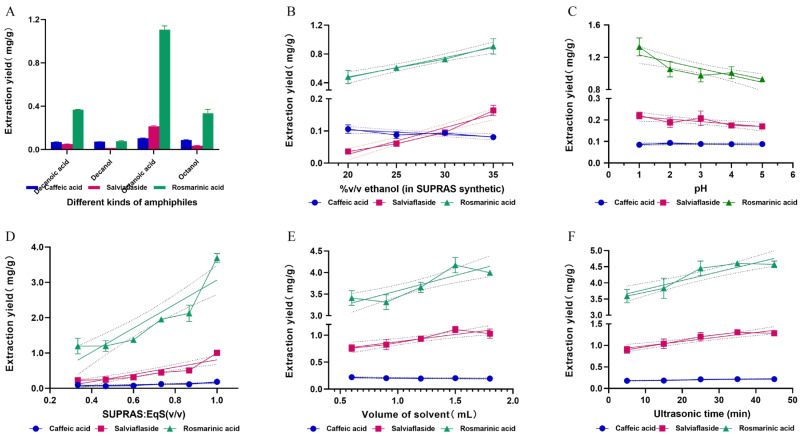
The impact of optimization factors on the yields of three phenolic compounds: various types of amphiphiles (**A**), percentages of ethanol (**B**), pH levels (**C**), varying ratios of SUPRAS to EqS (**D**), different sample-to-solvent ratios (**E**), and ultrasonic extraction time (**F**).

**Figure 4 molecules-29-02067-f004:**
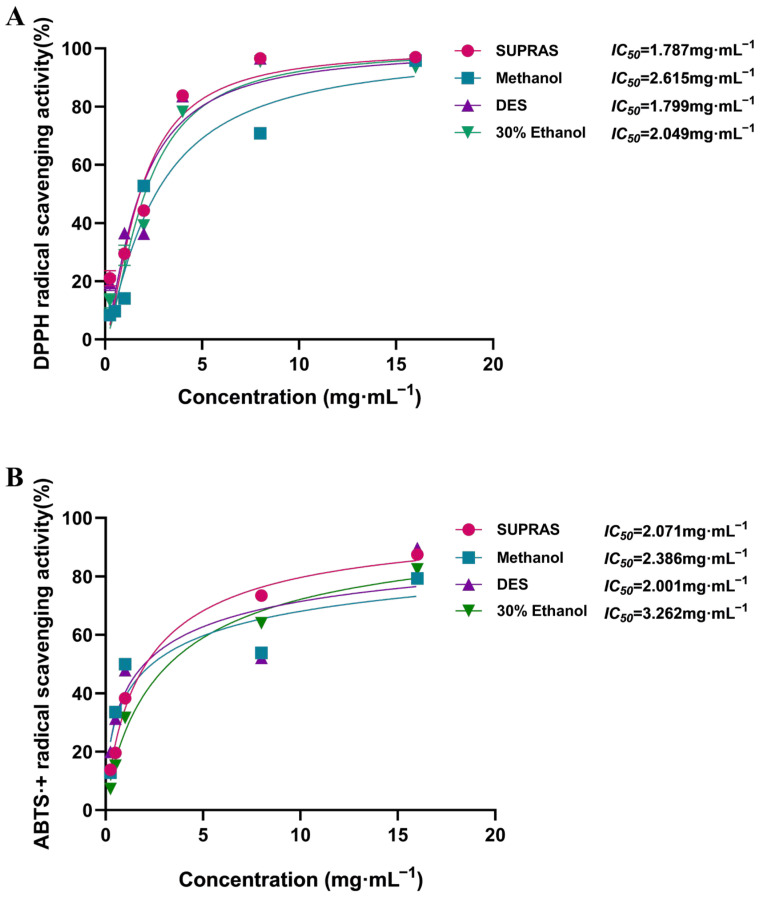
Comparative results of the antioxidant capacities of the extracts obtained by different extraction methods (DPPH method (**A**) and ABTS•+ method (**B**)).

**Figure 5 molecules-29-02067-f005:**
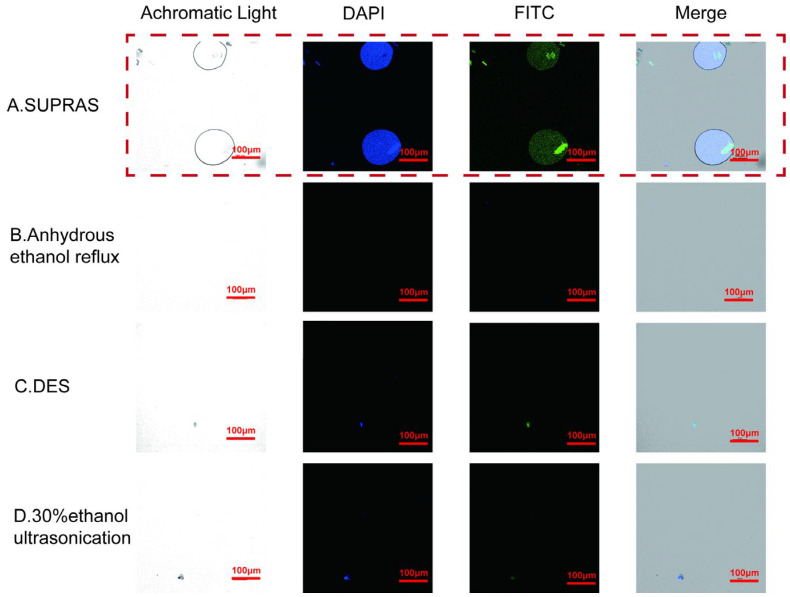
CLSM images displaying extracts from SUPRAS (**A**) compared to extracts obtained through alternative extraction techniques (**B**–**D**) are shown. The DAPI images were stimulated at 405.0 nm and emitted at 450.0 nm (scale bar, 100 μm).

**Figure 6 molecules-29-02067-f006:**
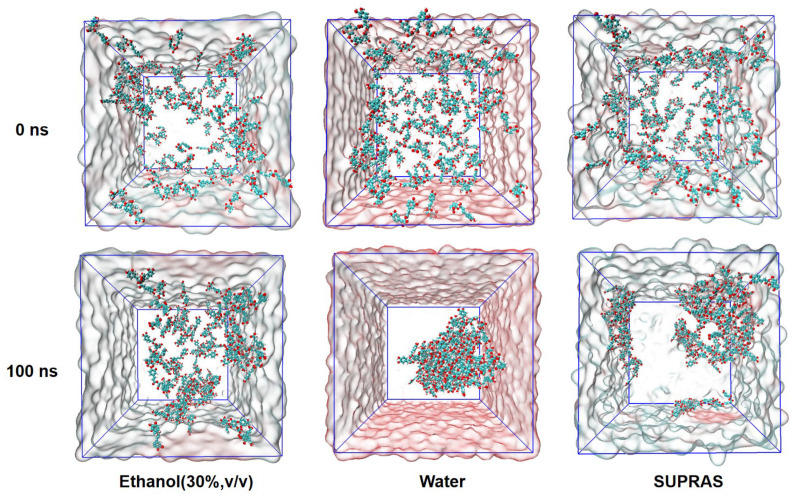
The distribution of rosmarinic acid in different solvent systems (SUPRAS, 70% *v*/*v* ethanol, and water) at 0 ns and 100 ns analyzed by molecular dynamics simulation.

**Figure 7 molecules-29-02067-f007:**
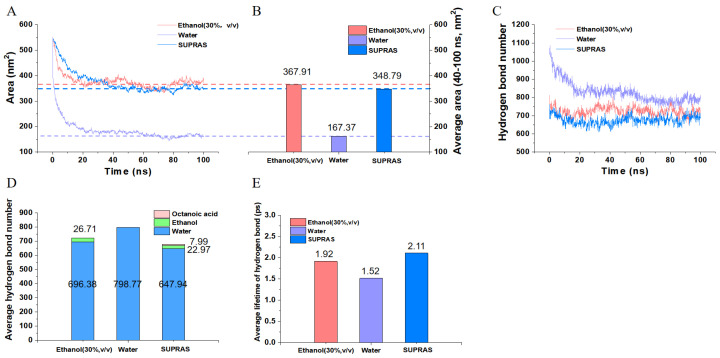
The simulation parameters of different extraction processes of rosmarinic acid. Solvent accessible surface area (SASA) from 0 ns to 100 ns and the corresponding average SASA (**A**,**B**) and the number (**C**,**D**) and average lifetime (**E**) of hydrogen bonds between rosmarinic acid molecules and different solvent systems.

**Figure 8 molecules-29-02067-f008:**
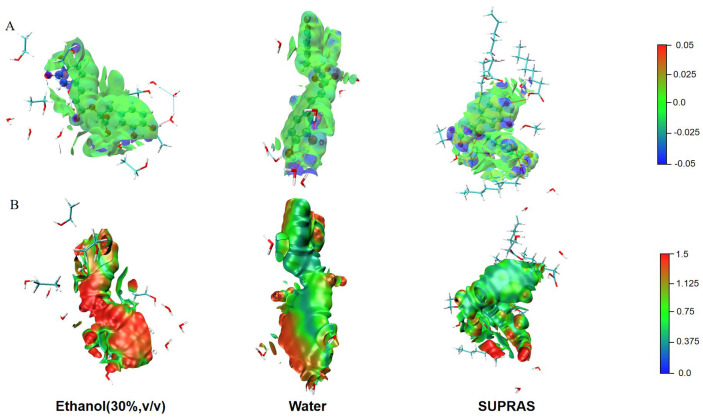
The simulation diagrams of average non-covalent interaction (aNCI) analysis (**A**) and thermal fluctuation index (TFI) analysis (**B**) for the extraction processes of rosmarinic acid.

**Table 1 molecules-29-02067-t001:** The results of analytical method validation.

Analyte	Linear Range	r^2^	RSD (%)	Average Recovery (%)	Average Recovery RSD (%)	LOD (mg·mL^−1^)	LOQ (mg·mL^−1^)
Caffeic acid	y = 54,280x + 25.135	1.000	3.31	101.98	1.02	0.16	0.53
Salviaflaside	y = 16,218x + 14.32	0.9996	3.14	101.20	1.39	0.08	0.29
Rosmarinic acid	y = 28,587x − 11.364	0.9996	1.15	100.81	1.22	0.10	0.34

**Table 2 molecules-29-02067-t002:** Extraction of phenolic acids from PV in SUPRAS extracts compared with other methods and solvents.

Solvent	Extraction Yield (mg/g)				Ref
	Caffeic Acid	Salviaflaside	Rosmarinic Acid	Total	
SUPRAS	0.240 ± 0.009	1.443 ± 0.010	5.254 ± 0.057	6.937 ± 0.067	
30% ethanol	0.146 ± 0.027	0.800 ± 0.047	2.957 ± 0.001	3.903 ± 0.074	[33]
Anhydrous methanol	0.244 ± 0.002	1.357 ± 0.063	4.667 ± 0.109	6.468 ± 0.173	[10]
36% vol. water in ChCl/ethylene glycol	0.213 ± 0.011	1.199 ± 0.005	3.706 ± 0.046	5.117 ± 0.052	[34]
F	22.151	10.868	595.073	177.245	
P	0.003	0.012	0.000	0.000	

## Data Availability

The original contributions presented in the study are included in the article, further inquiries can be directed to the corresponding authors.

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
