# Peer review of "Green and Efficient Extraction of Phenolic Components from Plants with Supramolecular Solvents: Experimental and Theoretical Studies"

_molecules, 2024, doi:10.3390/molecules29092067_

Round 1

Reviewer 1 Report

Comments and Suggestions for Authors

The paper by Xia et al. investigates modern extraction methods of phenolic components from plants. In my opinion the work is interesting, and the data is adequate. However, there are some major concerns.

Specific comments

1)               Please enhance the resolution of the Figures

2)               The novelty of the work should also be highlighted in the introduction section.  Please specify clearly how the data can contribute to the existing literature.

3)               Please revise the manuscript for typing errors

4)               A more detailed chemical analysis of the extracts should be performed. The extracts contain a complex mixture of compounds and not only three (caffeic acid, salviaflaside, and rosmarinic acid)

5)               Determination of total phenols should be performed.

6)               After the above experiments the discussion should be revised

Comments on the Quality of English Language

Minor editing of English language required

Author Response

Q1. Please enhance the resolution of the Figures.

Answer: I apologize for the appearance of mistakes like the lack of clarity of the figures in the article. We re-exported the figures, ensuring the line charts were in 600dpi, tif format, and the others in 300dpi, tif format. In addition, we will provide a separate file for each image to enhance its clarity.

Q2. The novelty of the work should also be highlighted in the introduction section.  Please specify clearly how the data can contribute to the existing literature.

Answer: The innovative aspects of the supramolecular solvents-based extraction method and its significance for existing research have been added in the introduction. Importantly, the extraction efficiency and antioxidant activity of SUPRAS extracts were compared with those of the traditional methods, showing that the SUPRAS extraction method has higher extraction efficiency, more convenient extraction conditions and preparation methods and better antioxidant capacity, which has a broad application prospect. In addition, laser confocal microscopy photography and molecular dynamics simulation can further provide a theoretical basis for the excellent interaction mechanism of supramolecular solvents, including a larger solvent contact surface area, more types of hydrogen bonding between the extractant and supramolecular solvents, and stronger and more stable interaction force in SUPRAS.

Q3. Please revise the manuscript for typing errors

Answer: I am sorry for the careless expression. We have rechecked the entire text and corrected the spelling mistakes and grammatical errors that existed.

Q4. A more detailed chemical analysis of the extracts should be performed. The extracts contain a complex mixture of compounds and not only three (caffeic acid, salviaflaside, and rosmarinic acid).

Answer: Thank you for your meaningful questions. We have carefully considered this problem and will address it with you here, hoping to get your approval. Firstly, the HPLC method for the identification of phenolic acid extracts of Prunella vulgaris was based on the previous study by our group, which was able to isolate the main phenolic acids contained in the fruits of Prunella vulgaris[1]. Secondly, the study of Liu et al. proved that in the determination of the content of 16 phenolic compounds of Prunella vulgaris, rosemarinic acid and salviaflaside had the highest content, while caffeic acid came in second but had a substantially greater amount than the other compounds. Furthermore, recent studies have shown that the indicators that can be determined are mostly concentrated in a few compounds such as rosmarinic acid, salvinorin, caffeic acid. Therefore, the supramolecular solvent prepared in this study can efficiently extract the phenolic acids of Prunella vulgaris and does not contain too many other complex components, which is convenient and efficient and does not require complex separation methods[2]. In addition, according to the existing research, the plant has shown both in-vitro and in-vivo antioxidant potential this has been attributed to phenolic compounds such as caffeic acid, rosmarinic acid, rutin, and quercetin. The present experiment mainly focuses on phenolic acids, and these three compounds, caffeic acid, salviaflaside, and rosmarinic acid, have strong antioxidant capacity and have a much higher application value than other trace phenolic acids. Therefore, we selected these three compounds with comprehensive consideration [3].

[1] Xia B, Yan D, Bai Y, et al. Determination of phenolic acids in Prunella vulgaris L.: a safe and green extraction method using alcohol-based deep eutectic solvents[J]. Analytical Methods, 2015, 7(21): 9354-9364.

[2]Min, L. U. O., Ting, H. E., Lei, G. O. N. G., & Rongxiang, C. H. E. N. (2023). Determination of 16 Phenolic Compounds in Prunella vulgaris and Analysis Their Correlation with Antioxidant Activity.

[3]Ahmad, Gazanfar, et al. "Phytochemical Analysis and Anti-inflammatory Activity of Various Extracts Obtained from Floral Spikes of PRUNELLA VULGARIS L." Jordan Journal of Pharmaceutical Sciences 13.1 (2020).

Q5.    Determination of total phenols should be performed.

Answer: Thanks for your careful review. We have added relevant results, and the content of phenolic acid in the extract is 72.44%.

Q6. After the above experiments the discussion should be revised

Answer: Thanks for your careful review. We have added relevant discussion content.

Reviewer 2 Report

Comments and Suggestions for Authors

Supramolecular solvents (SUPRAS) offer several advantages over traditional extraction methods in terms of efficiency and environmental impact: Efficiency: SUPRAS have been shown to have efficient extraction capacities for target compounds The unique properties of SUPRAS, such as self-assembly and coalescence, provide a wide variety of interactions for analytes, leading to high extraction efficiency . Molecular dynamics simulations have demonstrated that SUPRAS exhibit high extraction efficiency compared to other solvents Green and Sustainable: SUPRAS are considered green and safe solvents. They are based on alkanols/alkanoic acids, which are more environmentally friendly compared to traditional organic solvents . Additionally, SUPRAS have low toxicity and are biodegradable, making them a sustainable alternative Specificity and Selectivity: SUPRAS can be designed and tailored for specific extraction purposes, allowing for the extraction of target compounds with high selectivity This specificity can lead to improved extraction of bioactive substances from natural resources Reduced Energy Consumption: The use of SUPRAS in extraction processes can potentially reduce energy consumption due to their efficient extraction capabilities This can contribute to overall process sustainability and cost-effectiveness. Potential for Industrial Applications: While the industrial application of SUPRAS is still limited, ongoing research and development aim to address this limitation As more specific and suitable SUPRAS are designed and investigated, their potential for industrial use is expected to increase In summary, SUPRAS offer a promising alternative to traditional extraction methods by providing efficient extraction capabilities, environmental sustainability, and the potential for tailored and selective extraction processes. Further research and development in this field are likely to enhance the use of SUPRAS in various industries and applications. Molecular Dynamics Simulation: Molecular dynamics simulations were employed to analyze the intermolecular interactions between the extracted compounds and the extractants in SUPRAS. This theoretical approach helped in understanding the extraction mechanism and efficiency of SUPRAS. It is necessary to present the information obtained from this section; there are no interaction graphs, hydrogen bonds, hydrophobic forces, etc.

Author Response

Q1. It is necessary to present the information obtained from this section; there are no interaction graphs, hydrogen bonds, hydrophobic forces, etc.

Answer: We apologize for the omission of figures due to the lack of careful checking of the article. Interaction diagrams, hydrogen bonding, and hydrophobic force diagrams related to molecular dynamics simulations have been added to the article content as Figure 6-8.

Reviewer 3 Report

Comments and Suggestions for Authors

The manuscript presents green and efficient procedure of phenolic components extraction from plants with supramolecular solvents. 

The work is interesting; however, there is no novelty because it is a copy of the premise previously published in the Sustainable Chemistry and Pharmacy, Volume 38, April 2024, 101445, from the same Authors. This is also indicated by the 45% similarity indicated in the iThenticate report.

Caring about the quality of the work I review, I cannot recommend such behavior and am in favor of rejecting the work. 

I must admit that the original work published in Sustainable Chemistry and Pharmacy is very interesting. 

Author Response

Q1. The work is interesting; however, there is no novelty because it is a copy of the premise previously published in the Sustainable Chemistry and Pharmacy, Volume 38, April 2024, 101445, from the same Authors. This is also indicated by the 45% similarity indicated in the iThenticate report.

Answer: Thanks for your careful examination. Thank you for scrutinizing and bringing up this major issue. We deeply regret the overlap in ideas and content between the two articles and appreciate your observation. Recognizing the issue's gravity, we aim to provide a thorough explanation for the purpose of these experiments. Our study focuses on investigating the effectiveness and mechanism of supramolecular solvent extraction on various bioactive components using Prunella vulgaris plant samples. Our goal is to demonstrate the reliability and applicability of the supramolecular solvent extraction system. Various supramolecular solvents were synthesized by extracting a range of bioactive substances for validation purposes. Our future research aims to explore their potential applications with other bioactive ingredients like alkaloids and triterpenes, as well as with various medicinal plant samples.

Through a thorough revision of the article's content, the repetition rate was significantly reduced. The cited articles were carefully reviewed. This study on supramolecular solvents emphasizes their advantages in enhancing the antioxidant activity of extracts compared to traditional solvents. These findings highlight the potential benefits of using supramolecular solvents, offering new opportunities for researchers in this field. In addition, compared with flavonoids, the types and preparation conditions of supramolecular solvents are different in the process of extracting phenolic acids. We hope to let researchers see the differences in the extraction conditions of different types of compounds, and provide preliminary process guidance for further application. Finally, in the molecular dynamics simulation, the interaction force between supramolecular solvent and phenolic acid is also different, hoping that this difference can provide a new idea for the study of the mechanism of supramolecular solvent.

Thank you for highlighting this significant issue. This revision has emphasized the importance of innovation in the article. We will take your suggestion into consideration and ensure that similar mistakes are avoided in our future work.

Thank you very much for your comment on my paper.

Sincerely yours,

Round 2

Reviewer 1 Report

Comments and Suggestions for Authors

This version can be accepted

Comments on the Quality of English Language

Minor corrections are needed

Author Response

Thank you very much. We have tried our best to improve the English language.

Reviewer 3 Report

Comments and Suggestions for Authors

The work has been completely rewritten to get rid of the similarity of the text. This in no way changes the quality of the work and the similarity in content to the publication mentioned in the first review. Taking care of the quality of the published work, I cannot recommend this work. 

New developments should be published, not everything that can be done in the laboratory. Applying the method to successive samples of different plants does not generate scientific novelty. It is high time to stop publishing everything that can be written. 

Author Response

Thank you very much for your advice. However, according to the review of the literature, in the realm of bioactive compound research, it is a common practice to extract various types of bioactive compounds from a single plant sample. This is attributed to the intricate and diverse composition of plant components. Extracting different active compound components is crucial in enhancing their utilization. For instance, studies have focused on extracting diverse active compounds like polysaccharides and flavonoids from Polygonatum sibiricum using DES [1,2]. Similarly, Acanthopanax senticosus has been investigated for its flavonoids and polysaccharides extraction using DES [3,4]. Furthermore, research has explored the impact of green extraction solvents on the phenolic acids and flavonoids of propolis [5].

In research on green solvent extraction, it is essential to easily and quickly customize various types of deep eutectic solvents (DES), ionic liquids (IL), and supramolecular solvents (SUPRAS) for more efficient extraction of bioactive compounds with varying polarities [6,7,8]. This approach can broaden the use of innovative green solvents, address common challenges, and establish a systematic method for solvent application. Furthermore, existing studies highlight the simplicity of steps and convenience of synthesis conditions in optimizing the extraction process, as green solvents increasingly favor straightforward synthesis methods in their application.

Regarding supramolecular solvents, numerous studies have explored their potential in extracting bioactive compounds with varying polarities from plant samples. However, existing literature lacks a thorough comparison of the differences in supramolecules suitable for extracting compounds with different polarities from fixed samples [9]. Our study revealed that SUPRAS prepared with long chain carboxylic acids are more effective in extracting highly polar compounds. Increasing the ethanol proportion is more conducive to extracting phenolic acids compared to flavonoids. The pH of the poor solvent water, another key component in the synthesis system, significantly impacts extraction efficiency. Compounds with high polarity require low pH synthesis conditions, highlighting the importance of compound polarity in customizing SUPRAS. Furthermore, there is a notable difference in extraction ratios between different polar compounds using SUPRAS and EqS. Phenolic acids, with higher polarity, are better extracted with SUPRAS, while EqS is more efficient in extracting less polar compounds like flavonoids [10]. These findings aim to assist future researchers in quickly screening extraction conditions when utilizing SUPRAS for various compound types, ultimately reducing unnecessary workload.

This study aimed to explore the relationship between chemical structures of polar compounds like phenolic acids and flavonoids and the intermolecular forces within SUPRAS through molecular dynamics simulations to understand the extraction selectivity of SUPRAS for these compounds. Results indicated a higher number of hydrogen bonds in the octanoic acid SUPRAS solvent system, potentially crucial for the extraction of different polar compounds. Moreover, the average non-covalent interaction (aNCI) plot revealed more dark blue hydrogen bonding regions in the SUPRAS system for phenolic acids extraction, distinguishing it from the forces involved in flavonoids extraction. However, this exploration of extraction mechanisms is preliminary and lacks depth, suggesting the need for further investigation in future studies.

Reference: 1. Zhang, H., Hao, F., Yao, Z., Zhu, J., **g, X., & Wang, X. (2022). Efficient extraction of flavonoids from Polygonatum sibiricum using a deep eutectic solvent as a green extraction solvent. Microchemical Journal, 175, 107168.

  1. Meng, J., Guan, C., Chen, Q., Pang, X., Wang, H., Cui, X., ... & Zhang, X. (2023). Structural Characterization and Immunomodulatory Activity of Polysaccharides from Polygonatum sibiricum Prepared with Deep Eutectic Solvents. Journal of Food Quality, 2023.
  2. Xue, J., Su, J., Wang, X., Zhang, R., Li, X., Li, Y., ... & Chu, X. (2024). Eco-Friendly and Efficient Extraction of Polysaccharides from Acanthopanax senticosus by Ultrasound-Assisted Deep Eutectic Solvent. Molecules, 29(5), 942.
  3. Zhang, X., Su, J., Chu, X., & Wang, X. (2022). A green method of extracting and recovering flavonoids from Acanthopanax senticosus using deep eutectic solvents. Molecules, 27(3), 923.
  4. Kekeçoğlu, M., & Sorucu, A. (2021). Determination of The effect of green extraction solvents on the phenolic acids and flavonoids of propolis. Journal of Research in Veterinary Medicine, 41(1), 49-54.
  5. Ahmad, I., Yanuar, A., Mulia, K., & Mun'Im, A. (2018). Ionic liquid-based microwave-assisted extraction: Fast and green extraction method of secondary metabolites on medicinal plant. Pharmacognosy Reviews, 12(23).
  6. Dheyab, A. S., Abu Bakar, M. F., AlOmar, M., Sabran, S. F., Muhamad Hanafi, A. F., & Mohamad, A. (2021). Deep eutectic solvents (DESs) as green extraction media of beneficial bioactive phytochemicals. Separations, 8(10), 176.
  7. Sánchez-Vallejo, C., Ballesteros-Gómez, A., & Rubio, S. (2022). Tailoring composition and nanostructures in supramolecular solvents: Impact on the extraction efficiency of polyphenols from vegetal biomass. Separation and Purification Technology, 292, 120991.
  8. Ueda, K. M., Keiser, G. M., Leal, F. C., Farias, F. O., Igarashi-Mafra, L., & Mafra, M. R. (2024). A New Single-Step Approach Based on Supramolecular Solvents (SUPRAS) to Extract Bioactive Compounds with Different Polarities from Eugenia pyriformis Cambess (Uvaia) Pulp. Plant Foods for Human Nutrition, 1-8.
  9. Yu, Z. L., Hou, S. Y., Lin, L. M., Chu, Y. R., Li, Y. M., Zhang, Z. M., ... & **a, B. H. (2024). Experimental optimization and mechanism analysis of extracting flavonoids with the supramolecular solvents-based methods. Sustainable Chemistry and Pharmacy, 38, 101445.

Thank you very much for your comment on our paper.

Sincerely yours,
